# Association of Protein Intake during the Second Year of Life with Weight Gain-Related Outcomes in Childhood: A Systematic Review

**DOI:** 10.3390/nu13020583

**Published:** 2021-02-10

**Authors:** Natalia Ferré, Verónica Luque, Ricardo Closa-Monasterolo, Marta Zaragoza-Jordana, Mariona Gispert-Llauradó, Veit Grote, Berthold Koletzko, Joaquín Escribano

**Affiliations:** 1Pediatric Nutrition and Human Development Research Unit, Universitat Rovira i Virgili, 43201 Reus, Spain; natalia.ferre@urv.cat (N.F.); veronica.luque@urv.cat (V.L.); ricardo.closa@urv.cat (R.C.-M.); 2Institut d’Investigació Sanitaria Pere Virgili, 43001 Tarragona, Spain; marta.zaragoza@iispv.cat (M.Z.-J.); mariona.gispert@iispv.cat (M.G.-L.); 3Pediatrics Unit, Hospital Universitari de Tarragona Joan XXIII, 43005 Tarragona, Spain; 4Department Paediatrics, Dr. von Hauner Children’s Hospital, LMU University Hospital, Ludwig-Maximilians Universität München, 43201 Munich, Germany; veit.grote@med.lmu.de (V.G.); berthold.koletzko@med.lmu.de (B.K.); 5Pediatrics Unit, Hospital Universitari Sant Joan de Reus, 43204 Reus, Spain

**Keywords:** early protein intake, obesity risk, metabolic programming, systematic review

## Abstract

There is accumulating evidence that early protein intake is related with weight gain in childhood. However, the evidence is mostly limited to the first year of life, whereas the high-weight-gain-velocity period extends up to about 2 years of age. We aimed to investigate whether protein intake during the second year of life is associated with higher weight gain and obesity risk later in childhood. We conducted a systematic review with searches in both PubMed^®^/MEDLINE^®^ and the Cochrane Central Register of Controlled Trials. Ten studies that assessed a total of 46,170 children were identified. We found moderate-quality evidence of an association of protein intake during the second year of life with fat mass at 2 years and at 7 years. Effects on other outcomes such as body mass index (BMI), obesity risk, or adiposity rebound onset were inconclusive due to both heterogeneity and low evidence. We conclude that higher protein intakes during the second year of life are likely to increase fatness in childhood, but there is limited evidence regarding the association with other outcomes such as body mass index or change in adiposity rebound onset. Further well-designed and adequately powered clinical trials are needed since this issue has considerable public health relevance.

## 1. Introduction

The World Health Organization (WHO) recommends exclusive breastfeeding during the first six months of life followed by continued breastfeeding along with nutritious complementary foods up to the age of 2 years or beyond [1]. Human milk has a lower protein content compared with typical infant formulas [2,3]. The amount of protein and the amino acid profile in breast milk is specific and genetically regulated for each mammal species, adapted to its requirements [4,5]. Breastfed infants exhibit lower weight gain velocity and a subsequent lower obesity risk later in life compared with those fed a cow’s milk-based formula [6,7,8].

There is compelling evidence—from observational studies, as well as some randomized clinical trials (RTC)—demonstrating that a high protein intake during the first year of life is associated with higher body mass index (BMI), higher fat mass, as well as increased risk of overweight or obesity later in life [9,10,11,12]. An RCT designed with this purpose showed an effect of the lower-protein formula in reducing BMI at 2 years and later in childhood [8,13]. The systematic review conducted by Hörnell et al. concluded that protein intake during the last half of the first year is associated with enhanced growth and increased overweight risk later in childhood [14].

Both the quantity and the quality (i.e., whey:casein ratio and content of branched-chain amino acids) of protein may be important. It is hypothesized that higher protein intake stimulates growth and adipogenesis through the insulin growth factor 1 (IGF-1) axis and the mammalian target of rapamycin (mTOR) pathway [5,15,16,17]. Branched-chain amino acids, especially leucine, seem to play an important role [18]. The so-called early protein hypothesis was tested in an RCT [8], which showed a lower concentration of insulinogenic amino acids and a lesser IGF-1 axis activation in breastfed infants and in infants fed with a lower-protein-content formula (similar to that of human milk) compared with infants fed with a higher-protein-content formula [19].

When complementary feeding starts, usually towards the end of the first half year of life, dietary protein intake tends to progressively increase and generally surpasses the age-specific recommendations [20,21,22]. However, the shift between either full breastfeeding or formula feeding to a complete family diet is also spread over the second year of life. The protein intake during the second year of life usually exceeds by far the dietary intake recommendations to an even greater extent than during the first year [20,21,23]. Considering this fact, it is not inappropriate to wonder if this high protein intake during the second year could produce harmful effects later in life as happens with the protein intake during the first year. Moreover, the “programming” effect is mainly attributed to a critical window of sensitivity that is established in the first 1000 days of life, including fetal growth, and this period lasts up to the second year of life. The first two years of life are probably the critical window because it is a period with a high plasticity and growth velocity that could lay the basis for future development. Accordingly, there is broad evidence that rapid weight gain during the first two years of life exercises a modulatory effect on later obesity risk [24], but the most consistent data reporting a significant effect of protein intake on IGF-1 activation, fat mass gain, and later BMI are restricted to the diet during the first year of life [9,10].

Despite the WHO recommendation to breastfeed infants until the age of 2 years, there are several barriers to maintaining breastfeeding for long periods [25,26]. Although a considerable proportion of infants are breastfed at birth, breastfeeding rates fall quickly with increasing age, and only a relatively small proportion of infants are breastfed beyond 12 months [25,27,28]. Infants who are not breastfed at the age of one year often are given regular cow’s milk as a drink, which provides three times the protein supply compared to human milk. Therefore, it is of considerable practical relevance to explore whether high protein intakes during the second year of life could also be associated to growth and obesity risk later in childhood, as shown for protein supplies in infancy.

The aim of this systematic review is to evaluate the effects of protein intake during the second year of life on weight and fat mass gain and the subsequent risk of obesity development later in childhood.

## 2. Materials and Methods 

We used the PRISMA (Preferred Reporting Items for Systematic Reviews and Meta-Analyses) method to systematically review the articles that assessed the effects of protein intake during the second year of life on childhood health later in life. We conducted the review using the electronic database PubMed^®^/MEDLINE^®^ (https://www.ncbi.nlm.nih.gov/pubmed/) and the Cochrane Central Register of Controlled Trials (https://www.cochranelibrary.com/central) and by searching clinical trials (including RCTs or prospective cohort studies) or reviews published up to 30 May 2020. The search terms conducted in each database are presented in Appendix A. We selected all the studies in humans that fulfilled the aim (effects of protein intake during the second year of life on all outcomes related to weight gain and the implicated mechanisms) without outcome restriction but published in English or Spanish. Study population was restricted to healthy term infants that did not present special needs or altered growth (i.e., we discarded all the studies performed in preterm or low-birth-weight infants as well as all the interventions performed in ill patients). Studies performed in low-income areas (in which there might be a high undernutrition risk or interventions aimed to improve growth in those areas) were also excluded from the analysis. In addition to the studies obtained by database searches, reference lists from reviews, meta-analyses, and the retrieved articles were checked over to identify further relevant studies.

As described previously, our main objective was to assess the effects of the protein intake during the second year of life on excessive weight gain and obesity risk development later in childhood. Primary outcomes were as follows:All the fatness measurement approaches (like subcutaneous skinfolds (millimeters), % body fat (%BF), fat mass (FM) (kilograms), or fat mass index (FMI) (kilograms/meter^2^),Body mass index (BMI) (kilograms/meter^2^ or z-score), andRisk of excessive weight (overweight and obesity risks) (RR, OR, or frequency).

In addition, we also investigated other secondary outcomes:Weight gain velocity (grams/month, z-scores or those that classified as rapid or normal growth), andAdiposity rebound (age in months or BMI at the adiposity rebound onset).

When possible, the principal summary measures were reported as differences in means, RR, or.

Literature search was conducted by one author (NF). Questions or uncertainties were solved by discussion with other authors (JE, VL, and RC). The initial search consisted of screening titles and abstracts, and the second step consisted of reviewing full-text articles to confirm or discard the study selection.

The information extracted from each individual study was as follows: first author and year of publication, study design, population and sample size, protein intake assessment (methods and measures), assessed health outcomes, effects and comments, and quality of the evidence.

For each included study, the quality of the evidence was assessed following the SING method that classifies the evidence in eight grades (ranged from 1++ (corresponding to high-quality metanalyses, systematic reviews of RCTs, or RCTs with a very low risk of bias) to 4 (corresponding to expert opinions)) and according to the classification, establishes the study as one of four grades of recommendation from A (when there is at least one metanalysis, systematic review, or RCT rated as 1++ and directly applicable to the target population or a systematic review of RCTs or a body of evidence consisting principally of studies rated as 1+ directly applicable to the target population and demonstrating overall consistency of results) to D (when there is only evidence level 3 or 4 or extrapolated evidence from studies rated as 2+) [29]. This method was developed by the Scottish Intercollegiate Guidelines Network for the NHS in Scotland to improve the current system for grading guideline recommendations.

## 3. Results

A total of 3982 references were revealed after duplicate removal and screened. In addition, we obtained 42 manuscripts from the reference lists of reviews and studies, which were also assessed for inclusion. After reading the title and the abstract, a total of 3971 studies were excluded and 53 full-text items were assessed for inclusion. From those, 39 were finally excluded (Appendix A, Table A1) and 14 were included in the systematic review (Figure 1). Reasons for exclusion were mainly being a narrative review, not assessing the protein intake during the second year of life, or associating the protein intake with outcomes measured at the same time period instead of at later ones.

Table 1 summarizes the included studies, grouped by assessed outcomes. Although 14 references were included, those manuscripts provide data from nine different studies with several reported in more than one included publication. We only found published data from one RCT [30], while all other studies were observational cohorts or secondary analyses of an RCT designed with a different purpose. In addition, we detected one registered RCT that is still ongoing (TOMI trial, NTC02907502).

Most of the included studies addressed the effect of protein intake on different outcomes related with the risk of overweight, such as BMI, fat mass index (FMI), percentage of body fat (%BF), subcutaneous skinfolds (SbSK), rapid weight gain, adiposity rebound (AR), or risk of excessive weigh [30,31,32,33,34,35,36,37,38,39,41,42,44,45]. We did not find studies assessing the effect of protein intake during the second year of life on blood pressure or neurodevelopment.

Most of the studies assessed the protein intake through dietary diaries over 3 to 7 days, but these were performed at different timepoints during childhood between 12 and 24 months of life. Additionally, the studied health outcomes were assessed at different ages. Different authors analyzed protein intake data with different approaches: some studies addressed total protein intake whereas others differentiated between protein sources, and some studies used protein intake as continuous variable while others categorized subjects in groups of low vs. high intake or compared intervention groups according to the protein content of the formula in an RCT. All these differences made the comparisons between studies difficult and prevented us from performing a pooled quantitative study of the effect using a meta-analysis. Results are presented below, grouped per health outcome.

Wall et al. reported an RCT with low degree of bias, which corresponds to a quality of evidence of 1+. All the other included references were longitudinal prospective cohort studies, so the quality of evidence was considered as 2+ for most of them. The most relevant biases were found for attrition, as 5 out of the 13 cohort study references showed a moderate risk of bias, and another 5 of them a high risk of bias (loss to follow-ups over 30% or 55%, respectively). Most of the studies showed a low confusion bias, but two of them (from the same cohort) showed a high risk [35,44], and another study [37] a moderate risk, depending on the confounders that were missing in their analyses. Selection bias was low in almost all the studies. Some of the cohorts had specific characteristics that differed from the healthy term infant population, which was our main focus (i.e., TEDDY study in newborns with genetic diabetes risk [42] and Gemini study with relatively high proportion of preterms [39]). These differences hampered the extrapolation to the general population. For one study [31], neither selection criteria nor the study sample characteristics were reported in the publication. Therefore, we categorized the study as unknown risk of selection bias. Finally, all the studies showed a low risk of information bias except for one [41], in which risk of bias was classified as high. A summary of the biases assessed in all the included references is shown in the Appendix B, Table A2 and Table A3.

### 3.1. Effects of Protein Intake on Fatness

#### 3.1.1. Included Studies

Five publications based on three studies assessed the effect of protein intake on fat mass, measured as FMI, %BF, or subcutaneous skinfolds [30,31,32,33,34].

#### 3.1.2. Results

Wall et al. [30] performed an RCT in 160 healthy one-year-old infants that were randomly assigned to consume cow’s milk (3.1 g protein/100 mL, control) or a Growing up milk Lite (1.7 g protein/100 mL, intervention) until the age of 2 years (Wall 2019). Apart from the differences in protein content, the products included other differences as the control group was provided with whole pasteurized and homogenized cow milk (supplied as powder) and the intervention group consumed a cow’s milk-based Growing up milk fortified with micronutrients (including vitamin D and iron), probiotics, and prebiotics (i.e., a symbiotic). Results showed that the intervention significantly reduced FMI (0.045 kg/m^2^, *p* = 0.026) and %BF (2.09%, *p* = 0.047) vs. control at 2 years.

Percentage of body fat was also assessed by Gunther et al. in the DONALD Study cohort that followed 203 healthy term infants from 6 months to the age of 7 years [32,33]. In one of the references [32], the authors analyzed the protein intake from 6 months to 2 years and categorized infants as high (H) or low (L) protein intake, if they were above or below the median at each age, respectively. They added the effect of continuing in the H group during the second year (intakes either at 18 or 24 months or the mean of the two values). They analyzed the association between remaining or not in the H group of protein intake during the first 2 years (H-H vs. H-L). They observed that the H-H group had a significant higher (16%) fat mass compared with the H-L group at 7 years. In addition, the H-H group showed a twofold odds ratio for having overfatness at 7 years (2.28 (95% CI: 1.06, 4.88; *p* = 0.03)). In the same study, considering the protein intake as a continuous variable (% of energy) and splitting by type of protein (total, animal, plant, and dairy), Gunther et al. found no association between these intakes at 18 or 24 months with %BF at 7 years [33]. Additionally, children in the H-H group at 2 years had a slightly higher %BF (β = 0.67 ± 0.31, *p* = 0.03) at 5 years compared with children in the H-L group [34].

In a French Cohort (ELANCE Study), Rolland-Cachera et al. showed that protein intake at 2 years was directly correlated with increased subcutaneous skinfolds at 8 years in 112 healthy children (*r* = 0.20, *p* = 0.04) [31].

#### 3.1.3. Conclusions

In summary, results from one RCT and two cohort studies that gathered a total of 521 children point to an association between a high protein intake during the second year of life and higher fatness in childhood. However, the timing of the outcome assessment was very variable (from 2 to 7 years), and there were discordant results when the protein intake was analyzed separately according to the food source or evaluated as a continuous variable. Therefore, we conclude that there is moderate evidence (B, 1+/2+) for a direct association between total protein intake during the second year of life and body fat mass.

### 3.2. Effects of Total Protein Intake on BMI

#### 3.2.1. Included Studies 

The effect of protein intake during the second year of life on later BMI was evaluated in 10 publications including data from seven studies [31,32,33,34,35,36,37,38,39,41].

#### 3.2.2. Results

In the DONALD Study cohort, results showed that infants classified in the higher-protein group until 2 years (H-H group) presented significant higher BMI z-scores at 2 years (β = 0.36 ± 0.13, *p* = 0.005) [34] and BMI z-scores at 7 years (from 0.4 to 0.8 standard deviations (SD) depending on confounders adjustment) [32] compared with the H-L group, but was not associated with the BMI z-score trajectory between 2 and 5 years [34]. Assessing this association as a continuous variable (energy % from protein), protein intakes at 12 months modulated BMI z-score at 7 years, but protein intakes at 18 or 24 months did not [33].

The effect of protein intake during the second year of life on the BMI later in childhood was analyzed in a further six cohorts [31,35,36,37,38,39,41]. In five of them [31,36,38,39,41], the results showed a significant association of the protein intake on the BMI z-score measured at different ages (ranging from 4 to 11 years). On the other hand, Cowin et al., in a subsample of the ALSPAC Cohort study in UK (*n* = 389) [35], found no association between protein intake at 18 months on the BMI z-score at 2.5 years.

More recently, Morgen et al. analyzed the protein intake at 18 months, splitting by protein food source. Results showed that BMI z-score at 7 years increased 0.012 (95% CI: 0.003, 0.021; *p* = 0.007) for each 5 g/day increase in dairy protein intake. And BMI z-score at 7 and 11 years increased 0.01 SD (95%CI: 0.004–0.017, *p* = 0.003) and 0.013 SD (95%CI: 0.005–0.020, *p* = 0.002) for each 2 g/day increase in meat and fish protein intake, respectively [41].

In the CAPs cohort [37] (in which the authors observed an association between protein intake at 18 months and BMI z-score at 8 years), BMI z-score growth trajectories from birth to 11.5 years were constructed with the growth mixture model to classify children in one of the three more representative classes (normal growth, early persistent BMI increase, and late increase). Protein intake at 18 months did not differ between children in the different growth trajectory groups.

#### 3.2.3. Conclusions

We found three studies including a total of 962 children that discarded or did not demonstrate an association between protein intake during the second year of life and the later BMI z-score or BMI z-score trajectory [33,35,37]. The other seven references including 3285 children (including some manuscripts on the same cohort studies that showed no association in a different analysis) found higher protein intake during the second year of life was associated with a higher BMI z-score in childhood [31,32,34,36,38,39]. All of the studies included here were longitudinal cohorts. Since results were highly heterogeneous and some studies showed considerable risk of bias, we conclude that there was a weak evidence of this association (D, 2+/2−).

### 3.3. Effects of Total Protein Intake on Later Obesity Risk

#### 3.3.1. Included Studies

The association between protein intake and the risk to develop overweight or obesity was investigated in three cohorts (The DONALD cohort [32,34], the Gemini cohort [39], and the TEDDY Study [42]).

#### 3.3.2. Results

In the DONALD cohort, the risk of overweight was more than twofold higher at 7 years if protein intake was persistently high during the first 2 years of life [32]. In the same cohort, Karaolis-Danckert showed an increased risk of being overweight at 5 years (27% vs. 15%, *p* = 0.003) among children who had a rapid weight gain pattern from birth to 2 years [34]. Pimpin et al. showed, in a twins cohort (*n* = 2435), that protein intake (measured at a mean age of 21 months) was associated with a trend of increased OR of being overweight or obese at 3 years (OR: 1.10, 95%CI 0.99–1.22, *p* = 0.075), while this was not maintained at 5 years of age [39]. In the TEDDY study (a multicenter cohort of 5563 infants at risk of having diabetes), protein intake at 1–2 years was not associated with the overweight risk at 5.5 years [42].

#### 3.3.3. Conclusions

In three different well-conducted cohort studies with a total of 8247 children, no consistent effect of protein intake during the second year of life on the overweight risk in childhood was demonstrated. Discrepancies between results could be due to the different ages at the outcome assessment in different studies. So, we conclude that the evidence for this association is low (C, 2+).

### 3.4. Effects of Protein Intake on Rapid Weight Gain (Secondary Outcome)

#### 3.4.1. Included Studies

Two of the included studies assessed the effect of protein intake on weight gain [34,39].

#### 3.4.2. Results

Pimpin et al., in the Gemini cohort (*n* = 2435), showed that those children consuming, at a mean age of 21 months, more than 16.3% of total energy from proteins (which corresponded to the top of the two quintiles) had higher weight gain until the age of 5 years (0.330 kg (95%CI 0.182–0.478) compared with children with the lowest intakes) without different length growth [39].

Results from the DONALD cohort (*n* = 249) showed that having a sustained consumption of higher protein intake during the first 2 years of life (H-H group) was not significantly associated with the probability to be classified as performing rapid weight gain in the same period (defined as weight increases >0.67 SD) compared to the H-L group [34]. Children with rapid early growth (0–2 years) showed higher %BF (16.7% vs. 18.0%, *p* = 0.02) and a higher risk of increased fatness (17% vs. 7%, *p* = 0.02) at 5 years, but protein intake during the first 2 years was not modulating changes in %BF trajectories beyond the 2 years.

#### 3.4.3. Conclusions

Data from two cohorts showed inconsistent results and a low evidence (C, 2+) for an association of protein intake during the second year of life with the child weight gain velocity until the age of 5 years.

### 3.5. Effects of Total Protein Intake on Adiposity Rebound (Secondary Outcome)

#### 3.5.1. Included Studies

One of the proposed mechanisms derived from the early protein hypothesis is related with an increased growth velocity that could lead to an earlier adiposity rebound (AR). Three of the included studies investigated how the protein intake in the second year of life modulated the AR process. This outcome was evaluated as age at the AR onset, as well as BMI z-score at the AR onset [31,44,45].

#### 3.5.2. Results

The effects of protein intake on AR were first investigated by Rolland-Cachera et al. in 1995 [31]. In the ELANCE cohort, protein intake (% energy from protein) at 2 years was associated with an AR at younger age (*r* = −0.2, *p* = 0.02) and with a higher BMI increase after the AR, which led to a subsequent association between the protein intake at 2 years and BMI at 8 years (*r* = 0.22, *p* = 0.03)). Moreover, those children with an early AR (before 4 years) had consumed higher protein intakes at 2 years compared with those showing a late AR (after 8 years) (16.6 ± 2.1% vs. 14.9 ± 2.1%, *p* < 0.01).

Dorosty et al., in a subsample of the ALSPAC Study cohort (*n* = 889, approximately 10% of the total sample), classified the children in three groups according to their age at AR (very early (<43 months), early (between 44 and 61 months), and late (>61 months), and found these not related to different protein intakes at 18 months [44].

The DONALD Study also explored the effect of protein intake on the BMI at the adiposity rebound. In girls, protein intake at 12–18 months was associated with an increase in the BMI z-score at the adiposity rebound onset (BMI z-score −0.61 vs. −0.08, *p* = 0.01; at the lower and higher tertile of protein intake, respectively), whereas there was no association in boys. Additionally, there were no age differences at adiposity rebound [45].

#### 3.5.3. Conclusions

Results from three cohorts including 1314 children showed an uncertain effect of protein intake during the second year of life on the AR advancement, with considerable heterogeneity. The heterogeneity was probably increased due to differences in the timepoints of protein intake assessment (12–18 months, 18 months, or 2 years), and the fact that one study only considered intake at one timepoint that cannot be considered representative of the protein consumption during the whole year. Considering the risk of bias in the included studies, we conclude that the evidence of this uncertain association was low (C, 2+) but cannot be discarded.

## 4. Discussion 

There is high-quality evidence in the scientific literature indicating that higher protein intake during the first year of life has a lasting programming effect increasing later obesity risk [5,8,14,46,47,48]. For protein intake during the second year of life, our review shows some indications for possible lasting effects, but the available evidence is based primarily on observational studies and does not allow firm conclusions.

### 4.1. Effect of Protein Intake during the Second Year of Life on Body Composition

There was high-quality evidence for a higher body fatness induced by increased protein intake during the second year of life as a short-term effect (at age 2 years) provided by an RCT (Gumli trial) [30]. The results that support the increased body fatness in the short term and later in life (at 2 and 7 years, in the DONALD cohort study) were provided by moderate evidence and hold some heterogeneity [32,34]. An extended follow-up of the Gumli trial sample would be valuable to improve the evidence of a long-term effect, if any.

All in all, it seems that the adipogenic effect exerted by higher protein intake during the first 12 months [49,50] could take place during a longer period, until the age of 2 years.

### 4.2. Effect of Protein Intake during the Second Year of Life on Body Mass Index

Apart from body composition and fatness, an indirect assessment of obesity could be addressed by the BMI measurement. Even though BMI is not able to differentiate if the excessive weight comes from fat or fat-free mass, this index is a convenient low-cost rule of thumb used to categorize a person according to body weight.

Some cohorts were able to demonstrate a significant association between protein intake during the second year of life and BMI [31,32,36,38,39,41], however, the biases of most of the studies were substantial, and this reduced the certainty of the effect observed.

The DONALD study found an association of protein intake during the second year of life with BMI z-score at 2 and 7 years [32,34]. However, they could not confirm an association between protein intake at 2 years and BMI trajectories between 2 and 5 years. These results could suggest that changes exerted during the first two years of life remain permanent rather than programming a later change in weight gain velocity.

The study by Garden et al. reported an association between protein intake during the second year and BMI z-score at 8 years that was not maintained later at age 11.5 years [36,37]. This could be because many other lifestyle patterns influencing BMI could take place; and also because puberty changes occurring during this period actually take place at different rhythms. In fact, the lack of information about the puberty stage is a weakness of the study performed by Garden et al. that did not control for this confounding factor. At 8 years, most children are still prepuberal, and we do not foresee an interference with the observed results. However, at 11.5 years, differences in maturity are considerable, and this could partially hide a possible long-term effect of early protein intake, if any, that could reappear later.

In summary, available evidence from study cohorts suggests a possible association between protein intake during the second year of life and an increased BMI z-score later on. There was low evidence of this association due to heterogeneity and the considerable biases observed in the cohort studies.

### 4.3. Effect of Protein Intake during the Second Year of Life on Later Obesity Risk

Another important primary outcome of interest was the effect on excessive weight (either overweight or obesity) risk. Analyzing this outcome is an attempt to assess the clinical relevance of the possible induced effects, as it reflects a well-recognized pathologic condition rather than an association with a risk factor.

The DONALD cohort supported the hypothesis of an increased overweight risk by having a persistently higher protein intake during the first two years of life (compared to only the first year of life) [32]. The results from the Gemini cohort did not confirm this hypothesis. The apparently contradictory results could be due to several factors: the Gemini cohort was performed in twins and included also preterms, thus, possibly low-birth-weight infants with different growth patterns compared to our target population, which could hinder extrapolating conclusions. On the other hand, in the Gemini cohort, protein intake was assessed at a mean age of 21 months, within a range of 17–34 months. Thus, some of the intake data were obtained during the third year of life instead of during the second. Furthermore, these results were obtained at a unique timepoint, which could not be representative of what happened during the second year. However, the results from the DONALD study, obtained from two different timepoints and classifying children according to an apparently persistent high protein intake, could be more representative of the high-protein diet during a longer period.

### 4.4. Effect of Protein Intake during the Second Year of Life: Outcomes Potentially Related with the Mechanism Inducing Increased Obesity Risk

Finally, we reviewed the evidence on potential mechanisms through which protein intake could increase later BMI or obesity risk. Rolland-Cachera et al. suggested that protein intake during the first 2 years would accelerate weight gain velocity, resulting in an anticipated adiposity rebound (AR), which could lead to a greater BMI from then onwards [31], and a possible earlier onset of puberty [51]. Regarding the association between protein intake during the second year and the accelerated weight gain between 2 and 5 years, the results from the DONALD and Gemini cohorts provided contradictory results. This could be due to several reasons: on one hand, children from the Gemini cohort had a higher proportion of preterms and low-birth-weight infants that could have a different growth pattern and/or a different maturation rhythm. On the other hand, it is also possible that a higher protein intake during the second year could accelerate weight gain later on, but not enough to achieve the 0.67 SD threshold, which was the one used in the DONALD cohort study. A possible hypothesis could be that increased protein intake during the first two years of life may accelerate weight gain and the development of the adipose tissue during this period; the exerted changes remain during childhood, but do not modify changes in trajectories of weight gain and fat mass gain beyond that age. In any case, the available evidences were not able to confirm this hypothesis.

The anticipated AR as a consequence of increased protein intake during the second year of life could not be confirmed by Dorosty et al. [44] in the ALSPAC cohort. Gunther et al., in the DONALD cohort, found a higher BMI z-score at the AR onset in girls with higher protein intake compared with girls with lower protein intake (association not found in boys) [45]. The discrepancies in the effects of early protein intake between genders have been previously observed in other studies, and a possible explanation could be that the IGF-1 axis response to proteins in female infants could be stronger than that of male infants [52].

In summary, the available evidence could not consistently support the hypothesis that the protein intake during the second year of life could lead to earlier AR onset.

### 4.5. Limitations for the Present Systematic Review

The most relevant limitation of this systematic review is the lack of specific and well-designed RCTs investigating the effect of protein intake during the second year of life on later obesity risk. We only found one published RCT [30] in which differences between the tested study products were not only restricted to the protein content but also differed in other nutrients such as vitamin D and iron, probiotics, and prebiotics. Moreover, risk of bias from the observational studies we found was moderate or high in most of cases.

### 4.6. “Importance for Public Health: A Window for Prevention” 

The novelty of our systematic review is that we focus on the effect of protein intake during the second year of life, whereas previous reviews did not differentiate this period from the first year [14,53,54]. Thus, if protein intake during this period was associated to later obesity risk, there would be an additional reason to promote breastfeeding or formulas with a lower protein content beyond the first year of life. We found moderate evidence for the association between protein intake during the second year of life and increased body fatness at 2 years. Evidence supporting either an increased risk of overweight or overfatness later in life was inconclusive. Overall, the quality of the evidence on effects on overweight and obesity is limited and based on observational cohort studies. Firm conclusions may be derived from randomized controlled intervention trials. In the Cochrane database of registered clinical trials, an ongoing RCT is reported, with more than 1600 children already recruited as infants (TOMI trial, Clinical trials.gov, NTC02907502).

## Figures and Tables

**Figure 1 nutrients-13-00583-f001:**
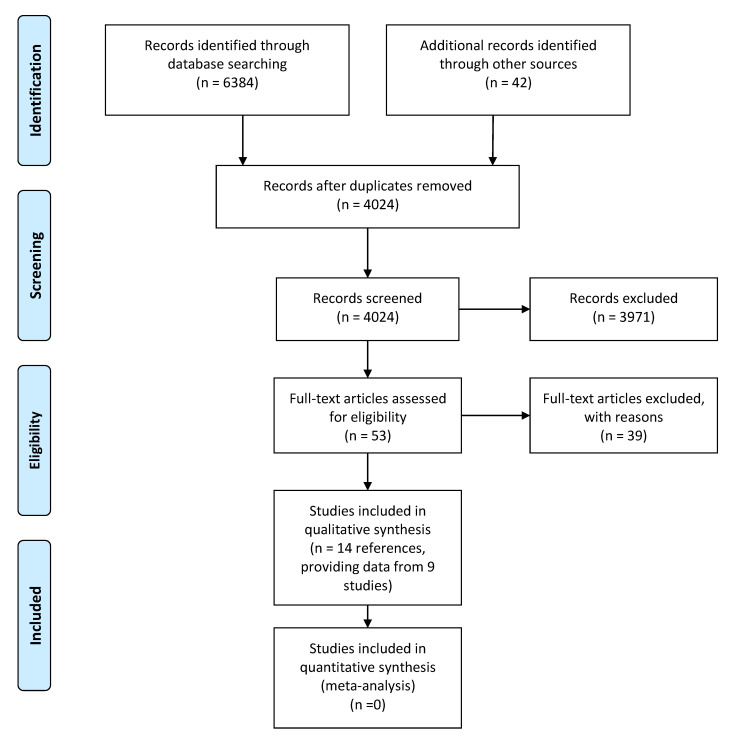
Preferred Reporting Items for Systematic Reviews and Meta-Analyses: The PRISMA Statement.

**Table 1 nutrients-13-00583-t001:** Included studies grouped by assessed outcomes.

Studies Assessing the Effect of Protein Intake during the Second Year of Life on Later Fatness in Childhood						
Reference	Study Design	Sample Population	Protein Intake Assessment	Assessed Outcomes	Effects and Comments	Study Quality
Wall 2019 [30]	RCT Gumli (NewZeland) Control: Cow’s milk (3.1 g protein/100 mL) Intervention: Growing up formula (1.7 g protein/100 mL) During second year of life	1 year ± 2 weeks healthy infants (>32S G) *n* = 160 allocated *n* = 134 analyzed	**Total protein** from 24 h recall and FFQ at 19, 20, 22, and 23 months. Mean of the protein intake from the four assessments	**BF** and **FMI** by bioimpedance at 1 and 2 years	After one year of intervention consuming a Growing up milk formula with lower protein content, the intervention group showed a reduction in FMI = 0.45 kg/m^2^ (*p* = 0.026) and a reduction in fat mass = 2.1% (*p* = 0.047) compared to the control group (cow’s milk) at 2 years of age.	1+ 17% drop-outs rate Low biases
Rolland-Cachera 1995 [31]	Cohort ELANCE Study (France)	Healthy infants recruited at 10 months of life *n* = 278 recruited *n* = 112 analyzed at 8 years	**Total protein** from interview (dietary history method) and 24 h recall at 2 years Protein intakes classified as high (H) (highest quartile), low (L) (lowest quartile), or medium (M) (second and third).	Anthropometry (BMI z-score and **skinfolds**) at 10 months, 2, 4, 6, and 8 years Age at adiposity rebound (AR): classified as early (<4 years) or late (>8 years)	Protein intake (% energy from proteins) at 2 years of age was directly correlated with subscapular skinfold at 8 years (*r* = 0.20, *p* = 0.04).	2− High drop-outs rate (59%), lack of information about inclusion and exclusion criteria or sample characteristics
Gunther 2007A [32]	Cohort DONALD (Germany)	Healthy infants (3–6 months at recruitment) *n* > 1100 recruited; *n* = 322 followed until the age of 7 years (singleton, term, and birth weight > 2500 g); *n* = 203 analyzed (all data available)	**Total protein** from 3-day weighed food diaries at 6, 12, 18, and 24 months. Protein intakes classified as high (H, highest 2 quartiles) or low (L, lowest 2 quartiles) in the first year (6 and 12 months) and then during the second year (12 plus 18–24 months). Comparison: H-L vs. H-H groups (high protein intakes sustained during 2 years vs. only the first year).	Anthropometry (BMI z-score, **%BF**, overweight and **overfatness risk**) at 7 years	The group with sustained higher protein intake during the first 2 years of life (H-H group) had 16% higher body fat at 7 years of age compared with the group with higher protein intake only during the first year (H-L group). The OR for overfatness at 7 years was 2.28 (95% CI: 1.06, 4.88; *p* = 0.03) in the H-H compared to the H-L group	2+ Low biases except for a moderate risk of follow-up bias
Gunther 2007B [33]	Cohort DONALD (Germany)	Healthy infants (3–6 months at recruitment) *n* > 1100 recruited; *n* = 322 followed until the age of 7 years (singleton, term, and birth weight >2500 g); *n* = 203 analyzed (all data available)	**Total, animal, plant, and dairy protein** intake from 3-day weighed food diaries at 6, 12, 18–24 months, 3–4 years, 5–6 years	Anthropometry (BMI z-score and **%BF**) at 7 years	Protein intake (total, animal, plant, or dairy protein) (% energy from proteins) during the second year of life was not associated with %BF at 7 years.	2+ Low biases except for a moderate risk of follow-up bias
Karolis-Danckert 2007 [34]	Cohort DONALD (Germany)	Healthy infants (3–6 months at recruitment) *n* > 1100 recruited; *n* = 408 followed until the age of 5 years (singleton, term, and birth weight >2500 g); *n* = 249 analyzed (all data available)	**Total protein** from 3-day food diaries at 12, 18, and 24 months. Protein intakes classified as high (H, highest 2 quartiles) or low (L, lowest 2 quartiles) for first year (12 months) and during the second year (12 plus 18–24 months). Comparison: L-L, L-H, H-L, and H-H (2 years of sustained high intakes vs. 2 years of low intakes vs. high intakes either the first or second year) groups.	Anthropometry (BMI z-score, **%BF**, **skinfolds**, **overfatness risk**, overweight risk, and rapid growth (0–2 y) (defined as weight z-score increases >0.67 change equivalent to a quartile increase))	Protein intake during the 2^nd^ year of life modulated BF% at 2 years (β = 0.67 ± 0.31, *p* = 0.03 for the H-H group compared to the H-L) but had no effect on the longitudinal change of %BF between 2 and 5 years. Children with rapid growth (0–2 years) exhibited thicker skinfolds, higher %BF (16.7% vs. 18.0%, *p* = 0.02), and higher overfatness risk at 5 years (17% vs. 7%, *p* = 0.02). However, linear mixed models showed that the association between rapid weight gain and %BF trajectories from 2 to 5 years was influenced by exclusive breastfeeding or %fat intake but not by protein intake during the second year of life.	2+ Low biases except for a moderate risk of follow-up bias
**Studies Assessing the Effect of Protein Intake during the Second Year of Life on Later BMI z-Score**						
**Reference**	**Study Design**	**Sample Population**	**Protein Intake Assessment**	**Assessed Outcomes**	**Effects and Comments**	**Study Quality**
Gunther 2007A [32]	Cohort DONALD (Germany)	Healthy infants (3–6 months at recruitment) *n* > 1100 recruited; *n* = 322 followed until the age of 7 years (singleton, term, and birth weight >2500 g); *n* = 203 analyzed (all data available)	**Total protein** from 3-day weighed food diaries at 6, 12, 18, and 24 months. Protein intakes classified as high (H, highest 2 quartiles) or low (L, lowest 2 quartiles) for first year (6 and 12 months) and during the second year (12 plus 18–24). Comparison: H-L vs. H-H groups.	Anthropometry (**BMI z-score**, %BF, and overweight risk) at 7 years	Sustained high protein intake during the first 2 years of life (H-H group) was associated with an increase of BMI z-score at 7 years (β = 0.37; *p* = 0.04) compared to having high protein intake only during the first year (H-L group).	2+ Low biases except for a moderate risk of follow-up bias
Gunther 2007B [33]	Cohort DONALD (Germany)	Healthy infants (3–6 months at recruitment) *n* > 1100 recruited; *n* = 322 followed until the age of 7 years (singleton, term, and birth weight >2500 g); *n* = 203 analyzed (all data available)	**Total, animal, plant, and dairy protein** intake from 3-day weighed food diaries at 6, 12, 18–24 months, 3–4 y, and 5–6 y	Anthropometry (**BMI z-score** and %BF) at 7 years	Protein intake (total, animal, plant, or dairy protein) (% energy from proteins) during the second year of life did not affect BMI-z-score at 7 years.	2+ Low biases except for a moderate risk of follow-up bias
Karolis-Danckert 2007 [34]	Cohort DONALD (Germany)	Healthy infants (3–6 months at recruitment) *n* > 1100 recruited; *n* = 408 followed until the age of 5 years (singleton, term, and birth weight >2500 g); *n* = 249 analzsed (all data available)	**Total protein** from 3-day weighed food diaries at 12, 18, and 24 months. Protein intakes classified as high (H, highest 2 quartiles) or low (L, lowest 2 quartiles) for first year (12 months) and during the second year (12 plus 18–24 months). Comparison: L-L, L-H, H-L, and H-H (2 years of sustained high intakes vs. 2 years of low intakes vs. high intakes either the first or second year) groups.	Anthropometry (**BMI z-score**, %BF, skinfolds, and overweight risk) at 5 years, rapid growth (0–2 y) (defined as weight z-score increases >0.67 (this change was is equivalent to a quartile increase))	Children with rapid growth (0–2 years) exhibited higher BMI z-score (−0.016 ± 0.99 vs. 0.41 ± 0.90, *p* < 0.001) at 5 years. However, the distribution of children with rapid growth was similar between groups of sustained or not high protein intakes during the first 2 years (H-H vs. H-L). Sustained higher protein intake during the first 2 years of life was modulating BMI z-score at 2 years (β = 0.36 ± 0.13, *p* = 0.005 for the H-H group compared to the H-L) but had no effect on the longitudinal change of BMI z-score between 2 and 5 years.	2+ Low biases except for a moderate risk of follow-up bias
Cowin 2001 [35]	Cohort ALSPAC (UK)	General population (recruited during pregnancy) *n* = 889 (≈10% randomly selected subsample from total sample); *n* = 389 analyzed (all data available)	**Total protein** from 3-day food diaries at 18 months	Anthropometry (BMI z-score) at 31 months	Protein intake at 18 months of age was not associated with changes in BMI z-score at 31 months but was associated with height (*r* = 0.176)	2− High drop-out rates (56%), adjustment for confounding factors not done.
Garden 2011 [36]	Cohort CAPS (Australia) Secondary analysis of an RCT (intervention: dust mite avoidance and omega-3 fatty acids supplementation)	Infants with asthma risk due to 1st degree relative’s affectation, recruited during pregnancy *n* = 362 (from total sample, *n* = 616)	**Total protein** from 3-day weighed food diaries at 18 months	Anthropometry (**BMI z-score** and waist circunference) at 8 years	Protein intake (g/day) at 18 months of age was associated with higher BMI z-score at 8 years (10 g/day of protein intake was associated to a BMI z-score increase in BMI ≈ 0.47 SD. Meat intake was also associated with BMI z-score and waist circumference at 8 years.	2+ Moderate risk of follow up bias
Garden 2012 [37]	Cohort CAPS (Australia) Secondary analysis of an RCT (intervention: dust mite avoidance and omega-3 fatty acid supplementation)	Infants with asthma risk due to 1^st^ degree relative’s affectation, recruited during pregnancy *n* = 616 recruited *n* = 370 analyzed	**Total protein** from 3-day weighed food diaries at 18 months	Anthropometry (**BMI**) at 1, 3, 6, 9, 12, 18, 24, 30, 36, and 42 months and 4, 5, 8, and 11.5 years → growth mixed model: classified in 3 sex-specific growth trajectories (normal, late increase, and early persistent increase)	Different BMI sex-specific growth trajectories (normal, early persistent increase, and late increase) till the age of 11.5 years were not associated with different protein intakes at 18 months of age.	2− High risk of follow-up bias (drop-outs = 52%), adjustment for some relevant confounders not done (i.e., smoking).
Ohlund 2010 [38]	Cohort Sweden	Healthy infants (6–18 months at recruitment) *n* = 300 recruited *n* = 127 analyzed at age 4 years	**Total protein** from 5-day food diaries at 12, 17–18 months, and 4 years	Anthropometry (**BMI z-score**) at 4 years Obesity risk at 4 years	Protein intake at age 17–18 months (gr/day and % from energy) was associated with changes in BMI z-score at 4 years (β = 0.05, *p* = 0.001 per each gr/day of protein intake or β = 0.14, *p* = 0.029 per 1% energy from protein intake).	2+ Low biases except for a high risk of follow-up bias (drop-outs = 66%)
Pimpin 2016 [39]	Cohort Gemini (twins, UK)	Twins cohort (recruited at 8.0 ± 2.2 months) *n* = 2435 recruited; *n* = 2154 analyzed The cohort included a 43.5% of preterms [40]	**Total protein** from 3-day food diaries between 17 and 34 months (mean 21 ± 1.2 months)	Anthropometry (**BMI z-score**) twice per year from 2 to 5 years Overweight risk at 3.5 years (normal, overweight, or obese) Linear mixed-effect models of growth trajectories from 2 to 5 years	Protein intake at age 21 months was associated with higher later BMI z-score (1% of energy from protein associated with BMI = 0.04 SD increase) between 3 and 5 years. Protein intake at 21 months in the top 2 quintiles (>16.3% from energy) was associated with increased BMI z-score between 3 and 5 years, compared with the lowest quintile (β = 0.323, 95%CI 0.115–0.531).	2++ Relative high cohort sample with low biases
Rolland-Cachera 1995 [31]	Cohort ELANCE Study (France)	Healthy infants Healthy infants recruited at 10 months of life *n* = 278 recruited; *n* = 222 followed until 4 years; *n* = 112 = analyzed at 8 years	**Total protein** from interview (dietary history method) at 2 years Protein intakes classified as high (H) (highest quartile), low (L) (lowest quartile), or medium (M) (second and third).	Anthropometry (**BMI z-score** and skinfolds) at 10 months, 2, 4, 6, and 8 years Age at adiposity rebound (AR)	Protein intake (% energy from proteins) at age 2 years was directly correlated with BMI at 8 years (*r* = 0.22, *p* = 0.03). Infants in the highest quintile of protein intake at 2 years showed an earlier AR and higher increase of BMI after 4 years, showing higher BMI at 8 years	2− High drop-outs rate (59%), lack of information about inclusion and exclusion criteria or sample characteristics
Morgen 2018 [41]	Cohort Danish National birth Cohort (DNBC) (Denmark)	General population (recruited during pregnancy) *n* = 36481 at 7 years *n* = 22047 at 11 years (from a total sample of 77,251)	**Total, meat/fish and dairy protein** from interview at 18 months	Anthropometry (**BMI z-score**) at 5, 12 months and 7, 11 years	Protein intake from dairy products at 18 months (per 5 g/day) increased BMI z-score at 7 years (β: 0.012, 95% CI: 0.003,0.021; *p* = 0.007) Protein intake from meat and fish at 18 months (per 2 g/day) increased BMI = 0.010SD (95% CI: 0.004, 0.017; *p* = 0.003) or 0.013 (95% CI: 0.005, 0.020; *p* = 0.002) at 7 and 11 years, respectively.	2− High drop-outs rate (72%). Very little information about dietary recording. Anthropometric data at 7 and 11 years reported by parents.
**Studies Assessing the Effect of Protein Intake during the Second Year of Life on Later Excessive Weight Risk**						
**Reference**	**Study Design**	**Sample Population**	**Protein Intake Assessment**	**Assessed Outcomes**	**Effects and Comments**	**Study Quality**
Gunther 2007A [32]	Cohort DONALD (Germany)	Healthy infants (3–6 months at recruitment) *n* > 1100 recruited; *n* = 322 followed until the age of 7 years (singleton, term, and birth weight >2500 g); *n* = 203 analyzed (all data available)	**Total protein** from 3-day weighed food diaries at 6, 12, 18, and 24 months. Protein intakes classified as high (H, highest 2 quartiles) or low (L, lowest 2 quartiles) for first year (6 and 12 months) and during the second year (12 plus 18–24). Comparison: H-L vs. H-H groups.	Anthropometry (BMI z-score, %BF, and **overweight risk**, according to ITOF criteria) at 7 years	Sustained high protein intake during the first 2 years of life (H-H group) was associated with a twofold increase of excessive weight at 7 years of age compared with having high intake only during the first year (H-L group). Overweight risk at 7 years, OR: 2.39 (95% CI: 1.14, 4.99; *p* = 0.02) in the H-H group compared to H-L	2+ Low biases except for a moderate risk of follow-up bias
Karolis-Danckert 2007 [34]	Cohort DONALD (Germany)	Healthy infants (3–6 months at recruitment) *n* > 1100 recruited; *n* = 408 followed until the age of 5 years (singleton, term, and birth weight >2500 g); *n* = 249 analyzed (all data available)	**Total protein** from 3-day weighed food diaries at 12, 18, and 24 months. Protein intakes classified as high (H, highest 2 quartiles) or low (L, lowest 2 quartiles) for first year (12 months) and during the second year (12 plus 18–24 months). Comparison: L-L, L-H, H-L, and H-H (2 years of sustained high intakes vs. 2 years of low intakes vs. high intakes in either the first or second year) groups.	Anthropometry (BMI z-score, %BF, skinfolds, **overfatness risk**, and overweight risk, according to ITOF criteria) at 5 years of age, rapid growth (0–2 years) (defined as weight z-score increases >0.67) (change equivalent to a quartile increase)	Children with rapid growth (0–2 years) had increased overweight risk (27% vs. 15%, *p* = 0.003) at 5 years compared to children without rapid growth.	2+ Low biases except for a moderate risk of follow-up bias
Pimpin 2016 [39]	Cohort Gemini (twins, UK)	Twins cohort (recruited at 8.0 ± 2.2 months) *n* = 2435 recruited; *n* = 2154 analyzed The cohort included 43.5% preterms [40]	**Total protein** from 3-day food diaries assessed at an age range between 17 and 34 months (mean 21 ± 1.2 months)	Anthropometry (BMI z-score) twice per year from 2 to 5 years **Overweight risk** at 3.5 years (normal, overweight, or obese, according to ITOF criteria) Linear mixed-effect models of growth trajectories from 2 to 5 years	Protein intake at 21 ± 1.2 months was associated with a trend of increased risk of overweight or obesity at 3 years (OR: 1.10, 95%CI 0.99–1.22, *p* = 0.075) that was not present at 5 years of age.	2++ Relative high cohort sample with low biases
Beyerlein 2017 [42]	Cohort TEDDY Study (Multicentric: USA, Germany, Finland, Sweden)	Healthy newborns with diabetes genetic risk by HLA screening *n* = 5563 (from a total *n* = 8676 enrolled) As described in bibliography, at 5 years a total of 80 children developed diabetes [43]	**Total protein** from 24 h recalls at 3 months or 3-day food diaries twice per year onwards (12, 18, 24, 30, 36 months, etc.)	Anthropometry (BMI z-score) quarterly from birth to 4 years and biannual from then onwards **Overweight** (BMI z-score >1 SD) and **Obesity** (BMI z-score >2 SD) **risks**	Protein intake during the second year of life (1–2 years) was not associated with changes in overweight or obesity risk at 5.5 years.	2+ Moderate risk due to 35% drop-outs rate
**Studies Assessing the Effect of Protein Intake during the Second Year of Life on Adiposity Rebound (Age and BMI in the Onset) (Secondary Outcome)**						
**Reference**	**Study Design**	**Sample Population**	**Protein Intake Assessment**	**Assessed Outcomes**	**Effects and Comments**	**Study Quality**
Rolland-Cachera 1995 [31]	Cohort ELANCE Study (France)	Healthy infants *n* = 278 recruited; *n* = 222 followed until 4 years; *n* = 112 analyzed at 8 years (from *n* = 222)	**Total protein** from interview (dietary history method) at 2 years Protein intakes classified as high (H) (highest quartile), low (L) (lowest quartile), or medium (M) (second and third).	Anthropometry (BMI z-score and skinfolds) at 10 months, 2, 4, 6, and 8 years **Age at adiposity rebound** (AR)	Protein intake (% energy from proteins) at 2 years of age was inversely correlated with age at the AR onset (*r* = −0.2, *p* = 0.02). Children with an early AR (before 4 years) had higher protein intake at 2 years compared with those children showing a late AR (after 8 years) (16.6 ± 2.1% vs. 14.9 ± 2.1%, *p* < 0.01). Infants in the highest quintile of protein intake at 2 years showed an earlier AR and higher increase of BMI after 4 years, showing higher BMI at 8 years	2- High drop-out rates (59%), lack of information about inclusion and exclusion criteria or sample characteristics
Dorosty 2000 [44]	Cohort ALSPAC (UK)	General population (recruited during pregnancy) *n* = 889 (randomly selected 10% subsample from total)	**Total protein** from 3-day food diaries at 8 and 18 months	Anthropometry (BMI z-score) from birth to 49 months **Adiposity rebound** (AR): classified as very early (<43 months), early (49–61 months) and late (>61 months)	Protein intake at 18 months was not significantly different in children classified in the different AR groups (according to age at AR onset). BMI before AR onset was similar in all the groups. These results suggested no effect of protein intake on age at AR onset or final BMI.	2+ Dietary recording plausibility assessment, adjustment for confounders and power calculation not done
Gunther 2006 [45]	Cohort DONALD (Germany)	Healthy infant (3–6 months at recruitment) *n* > 1100 recruited; *n* = 313 analyzed at 8 years (all data available)	**Total protein** from 3-day weighed food diaries at 12, 18, and 24 months	Anthropometry (BMI z-score): 3, 6, 9, 12, 18, 24, 36, and 48 months (and once yearly onwards) **Age at adiposity rebound** (AR) onset	Higher protein intake at 12–18 months was associated with higher BMI z-score at AR onset only in females. BMI z-score (internal z-score) was −0.11 and –0.66 in the upper and lowest terciles of protein intake at 12–18 months of age, respectively. Age at AR onset was independent of protein intake at 12–18 months in both genders.	2+ Unknown risk of follow-up bias because dropouts are not reported
**Studies Assessing the Effect of Protein Intake during the Second Year of Life on Weight Gain (Secondary Outcome)**						
**Reference**	**Study Design**	**Sample Population**	**Protein Intake Assessment**	**Assessed Outcomes**	**Effects and Comments**	**Study Quality**
Karolis-Danckert 2007 [34]	Cohort DONALD (Germany)	Healthy infants (3–6 months at recruitment) *n* = >1100 recruited; *n* = 408 (followed until the age of 5 years (singleton, term, and birth weight >2500 g); *n* = 249 analyzed at 5 years (dietary information available)	**Total protein** from 3-day food diaries at 12, 18, and 24 months. Protein intake classified as high (H, highest 2 quartiles) or low (L, lowest 2 quartiles) for first year (12 months) and then until the second year (12 plus 18–24 months). Results compared L-L, L-H, H-L, and H-H (2 years sustained high intake) groups.	Anthropometry (BMI z-score, %BF, skinfolds, and overfatness/overweight risk) at 5 years of age, **rapid growth (0–2 years)** (defined as weight z-score increases >0.67 (this change is equivalent to a quartile increase))	Distribution of children with rapid growth was similar in those infants consuming sustained higher protein intakes during the first 2 years (H-H) compared to those that did not (H-L). Linear mixed models analyses showed that the association between rapid weight gain and %BF trajectories up to 5 years was influenced by exclusive breastfeeding or %fat intake but not by protein intake during the second year of life. Protein intake during the second year of life was modulating %BF at 2 years (β = 0.67 ± 0.31, *p* = 0.03 for being from the H-H group compared to the H-L) but had no effect on the longitudinal change of %BF between 2 and 5 years.	2+ Low biases except for a moderate risk of follow-up bias
Pimpin 2016 [39]	Cohort Gemini (twins, UK)	Twins cohort (recruited at 8.0 ± 2.2 months) *n* = 2435 recruited; *n* = 2154 analyzed The cohort included a 43.5% of preterms [40]	**Total protein** from 3-day food diaries assessed at a range age between 17 and 34 months (mean 21 ± 1.2 months)	Anthropometry (BMI z-score) twice per year from 2 to 5 years Overweight risk at 3.5 years (normal, overweight, or obese) Linear mixed-effect models of **growth trajectories from 2 to 5 years**	Protein intake at 21 ± 1.2 months over 16.3% of total energy intake (in the top of the 2 quintiles) was associated with a greater weight gain up to 5 years of age (β = 0.330 kg, 95%CI 0.182–0.478 for the highest quintile vs. the lowest).	2++ Relative high cohort sample with low biases

FFQ: Food frequency quetionaire, BF: Body fat, FMI: fat mass index, BMI: body mass index, HLA: human leukocyte antigen, ITOF: International obesity Task Force.

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
