# Peer review of "Association of Protein Intake during the Second Year of Life with Weight Gain-Related Outcomes in Childhood: A Systematic Review"

_nutrients, 2021, doi:10.3390/nu13020583_

Round 1

Reviewer 1 Report

This is a very interesting systematic review, addressing the effect of protein intake during the second year of life in the obesity risk later in childhood, a topic not well studied and understood. Authors rightfully point that there is plenty of literature for the effect of protein intake during the first year of life and mainly during the first semester, due to the fact that breastmilk has lower protein content than formula.  Moreover, the sparse evidence and the discrepancy between the reported findings correctly guided the author not to perform a meta-analysis, but rather to procced with a systematic review of the literature. Their systematic review is very well contacted and the reported results are well documented. I question the inclusion of the two studies with study outcome puberty. Obesity in childhood is associated with premature signs of puberty such as the start of the acceleration of height growth but it cannot be considered as a weight gain-related outcome, especially due to the high intake in the 2nd year of life.  Their report might be highly confounded, as authors comment on Discussion, lines 477-490 and the association between premature puberty and obesity is very ambiguous. No consistent results are reported for boys, and sparce evidence for higher menarche rates for girls (1)    Authors should justify more the inclusion of these studies to their systematic review. Otherwise, they can be omitted without any significant loss to the manuscript. There are also some minor comments:

Introduction, line 57-58: Authors should rephrase this sentence and point the difference between developed and developing countries. The transition between full breastfeeding or formula to family diet takes place during the 2nd semester in developed countries, while in developing world this is very common practice.

Methods, line 120-122: Authors should provide more information about the SING method for the assessment of the quality of evidence.

Results, line 258: Please report if the initials SD stands for Standard Deviation. Moreover, insert SD next to “0.013”

Discussion, line 410-412: Their conclusion about the adipogenic effect of higher protein intake during the second year and higher body fatness is based only in the finding of a RCT and at the end of the period. Thus, no adipogenic effect is evident for the later childhood years. Therefore, authors should reconsider their conclusion about this outcome/

Discussion, line 432-437: Authors should rephrase, because the meaning of the sentence is unclear. The lack of association between protein intake during the 2nd year and BMI at 11,5 years could not be attributed to puberty changes. If in the study by Garden et all, the confounding effect of lifestyle or puberty stage were not considered, their results are highly confounding and the study is of low quality, which questions the quality of the result of the study at 8 years.

  1. Li W, Liu Q, Deng X, Chen Y, Liu S, Story M. Association between Obesity and Puberty Timing: A Systematic Review and Meta-Analysis. Int J Environ Res Public Health. 2017 Oct 24;14(10):1266. doi: 10.3390/ijerph14101266. PMID: 29064384; PMCID: PMC5664767.

Author Response

Dear Reviewer,

Thank you for all your comments that help to imporve our manuscript understanding. A point by point answer to your comments is included for your revision.

Yours faithfully,

Natalia Ferré

Reviewer 2 Report

This review systematically compares and analyzes previous studies on the effects of protein intake and type in 2-year-olds on subsequent childhood growth, especially obesity, body mass index, and other related biological changes. Please check the following, although they are minor points.

I think it makes sense that the authors are focusing on protein intake in 2-year-old children. However, it does not fully explain why the authors focused on 2-year-olds. It would be good if there was an explanation in the introduction etc. about what the meaning of a 2-year-old child in the growth process of childhood.

As a very minor descriptive point, make sure that the name of the researcher on line 43 is different from the original spelled.

Regarding the parentheses on lines 60 and 67, the types are different on the left side and the right side, so I think that the left side needs to be corrected.

Author Response

Dear Reviewer,

Thank you for your comments that help to imporve our manuscript. A point by point answer is included for your revision.

Yours faithfully,

Natalia Ferré
